# Phosphorylation regulates the binding of intrinsically disordered proteins via a flexible conformation selection mechanism

Na Liu[1,2,4], Yue Guo[1,3,4], Shangbo Ning[1,2] & Mojie Duan 🄳 [1✉]

Phosphorylation is one of the most common post-translational modifications. The phosphorylation of the kinase-inducible domain (KID), which is an intrinsically disordered protein (IDP), promotes the folding of KID and binding with the KID-interacting domain (KIX). However, the regulation mechanism of the phosphorylation on KID is still elusive. In this study, the structural ensembles and binding process of pKID and KIX are studied by all-atom enhanced sampling technologies. The results show that more hydrophobic interactions are formed in pKID, which promote the formation of the special hydrophobic residue cluster (HRC). The pre-formed HRC promotes binding to the correct sites of KIX and further lead the folding of pKID. Consequently, a flexible conformational selection model is proposed to describe the binding and folding process of intrinsically disordered proteins. The binding mechanism revealed in this work provides new insights into the dynamic interactions and phosphorylation regulation of proteins.

[1] Key Laboratory of magnetic Resonance in Biological Systems, State Key Laboratory of Magnetic Resonance and Atomic and Molecular Physics, National Center for Magnetic Resonance in Wuhan, Wuhan Institute of Physics and Mathematics, Chinese Academy of Sciences, Wuhan 430071, People's Republic of China. [2] School of biological and pharmaceutical engineering, Wuhan Polytechnic University, Wuhan 430023, People's Republic of China. [3] University of Chinese Academy of Sciences, Beijing 100049, People's Republic of China. [4] These authors contributed equally: Na Liu, Yue Guo. ✉email: mjduan@wipm.ac.cn

As one of the most common post-translational modifications (PTMs), phosphorylation is important in regulating protein synthesis, cell cycle, growth, apoptosis, cell division, and signal transduction[1–3]. Most of the phosphorylation sites are located in the intrinsically disordered proteins (IDPs) or intrinsically disordered regions[4,5]. The disordered nature of these proteins or regions facilitate the access for post-translational modification. Meanwhile, the functions of IDPs are regulated by the phosphorylation[6,7]. However, the detailed mechanism about the phosphorylation regulation on the structures and the interactions of IDPs remain elusive, which greatly impede the understanding of IDP function and the searching for "druggable" IDP targets[5,8].

In many cases, IDPs fold into the ordered structures upon binding to their function partner, which is termed as the "coupled folding and binding process"[9–13]. The kinase-inducible domain (KID) is a typical example which carry out its biological function through coupled binding and folding mechanism. As a part of the cAMP-response element binding protein (CREB), KID function as an inducible transcriptional activator[14,15]. Upon phosphorylation at Ser133, the phosphorylated KID (pKID) contacts with the KIX domain on the transcriptional coactivator CREB-binding protein (CBP) and modulates the target gene expression[16,17]. Two transient helices are present on the free-state pKID, i.e., $\alpha_A$ (from residues 120 to 129) and $\alpha_B$ (from residues 132 to 144), however, the $\alpha_A$ and $\alpha_B$ regions would fold into stable helical structures in the KIX-bound state[16,18].

The phosphorylation on residue Ser133 would increase the binding affinity of KID and KIX by almost two orders of magnitude[17,19–21]. Several models have been proposed to describe the phosphorylation modulation mechanism on the binding of pKID and KIX. One of them suggested that phosphorylation on Ser133 could shifts the conformation ensemble of pKID toward the configuration similar to the structure in pKID-KIX complex[18,22]. On the other side, many researchers believed that the increased affinity of pKID is induced by the intermolecular interactions related to the di-anionic phosphate group[17,23]. Another model proposed that the phosphorylation restrict the flexibility of the loop region on KID and therefore reduce the entropic cost for KIX binding[24]. Although many progresses have been achieved, the detailed mechanism of phosphorylation regulation on the binding process of pKID and KIX remains obscure.

It is great challenge to determine the structures of the intermediate states and the binding process due to the short lifetime of the intermediates[25]. One major debate about the pKID-KIX binding is the order of $\alpha_A$ and $\alpha_B$ folding in the intermediates and binding process. A recent kinetic experiment shows that the binding and folding of $\alpha_B$ region is prior to the binding and folding of $\alpha_A$ region[26]. Most of the large $\Phi$-value residues were located on the $\alpha_B$ region, indicate that the native contacts or binding of this region were completed in the transition state[26]. Nevertheless, the NMR researches show that the chemical shift values of the residues in the C-terminus of $\alpha_A$ (residues 128–132) in the intermediates are close to the bound state, which means the C-terminus of $\alpha_A$ are almost folded in the intermediate. On the other side, the chemical shifts differences of the $\alpha_B$ region in the intermediates and bound state are larger than the $\alpha_A$ region, indicate the $\alpha_B$ region are less folding than $\alpha_A$ in the intermediates[27]. To elucidate the above controversies requires the characterizing the binding intermediates and binding pathways.

In this work, the structure properties of free pKID or KID and the binding free energy surface of pKID/KID and KIX were characterized based on the enhanced sampling simulations. The results show that both free pKID and KID are mainly disordered with some transient helical structures on them. The secondary structure compositions of the pKID and KID are basically the same to each other, however, more long-range residue–residue interactions were observed in the pKID. The contacts between the hydrophobic residues on pKID would form special hydrophobic residue cluster (HRC). Although both of the KID and pKID would form encounter complex with KIX, only the structures with HRC which might pre-formed in free pKID would binding to the correct binding sites on KIX and leading to the folding and binding of pKID to form final complex. We proposed that the binding mechanism of the intrinsically disordered pKID would follow a flexible conformational selection mechanism.

## Results

**Structure ensembles of free pKID and KID**. PTMetaD-WTE method was employed to obtain the structure ensembles of free pKID and KID in solution. The quality of the simulated structure ensembles was evaluated by the prediction accuracy of secondary chemical shifts ($\delta_{cs}$). As shown in Supplementary Fig. 1, the predicted chemical shifts are agree well with the NMR measurements[18]. The RMSE of C$\alpha$ $\delta_{cs}$ between simulated and experimental results is 0.44 ppm for pKID and 0.47 ppm for KID. The RMSE of H$\alpha$ chemical shift for pKID and KID are 0.08 ppm and 0.06 ppm, respectively. The RMSE values are close to the system errors of the chemical shift calculating tool[28]. The results indicate that the structure ensembles obtained in our work provide a reasonable description of the structural properties of the IDPs. Based on the reliable structure ensemble, we found that both free pKID and KID are mainly disordered in solution, however, some transient helical structures were observed on the $\alpha_A$ (residues 120–129) and $\alpha_B$ (residues 134–144). The helical propensity of $\alpha_A$ and $\alpha_B$ were given in Fig. 1a. The N-terminal of KID and pKID have higher helical propensity than those on C-terminus. The helicity of $\alpha_A$ are about 50% in both pKID and KID. The average residue-helicity on $\alpha_B$ region of pKID (18.9%) is slightly higher than that on KID (14.6%), which is consistent with the experimental observations (Supplementary Table 1)[18].

Although the secondary structure compositions in pKID and KID are similar to each other, some obvious differences were observed on their tertiary structures. Based on the residue-residue contact maps (Fig. 1b), more interactions between two terminals (marked by black circle in Fig. 1b) were observed in pKID. The larger amount of residue contact probability indicates more compact structures formed after the phosphorylation. The hydrophobic residues (Leu128, Tyr134, Leu137, Leu138) around the pSer133 in pKID are more likely to form hydrophobic interactions than in the KID. The spatial closed hydrophobic residues would form a HRC. The contact number between the side-chain heavy atoms on the residues Leu128, Tyr134, Leu137, and Leu138 were calculated to quantitatively define the formation of HRC. The conformations with contact number larger than 15 are defined to be HRC structures. As can be seen in Fig. 1c, the HRC structures present on more than 40% conformations of free pKID. On the other side, almost no HRC formed on free KID as the probability of conformations with contact number larger than 8 is close to zero.

**The binding free energy landscape of pKID-KIX**. To characterize the binding and folding mechanism of pKID and KIX, the free energy landscapes (FEL) along the reaction coordinates were constructed. Figure 2 gives the free energy landscape as a function of the center-of-mass (COM) distances and the number of native contacts between pKID and KIX. Sugase et al.[27] proposed that the folding and binding process of pKID-KIX can be described by four-site exchange model, i.e., the free state, the encounter complex, the folding intermediates, and the bound state. All the four stages were clearly present on our simulated

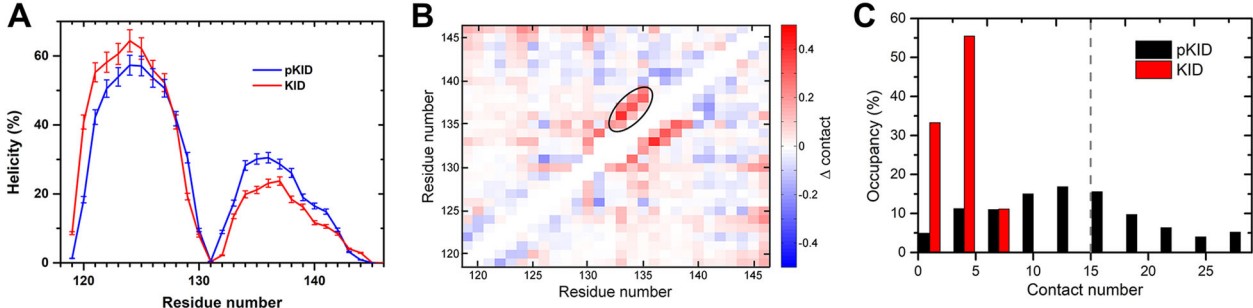

**Fig. 1 Structure properties comparison of free KID and pKID. a** The helicity of free pKID and KID. The errors are corresponding to the helicity fluctuation in the last 100 ns simulations, the helicity of residues were calculated from the initial time to different ending time, i.e., 220 ns, 240 ns, 260 ns, 280 ns and 300 ns. The standard deviations of the helicity are given as the error bars. **b** The intra-chain contact probability difference between the free pKID and KID. The contacts are calculated between the side-chain heavy atoms of the residues on the protein. Two residues are defined to be in contact if the distance between any two heavy atoms in different residues is <4.5 Å. Δcontact equals to the contact probability in pKID minus contact probability in KID. Red grid represents the residue-residue contact probability in pKID lager than that in KID, and blue grid represents the contact probability in KID lager than that in pKID. **c** Distributions of number of hydrophobic residue contact atoms in free pKID and KID. The hydrophobic residues include residues L128, Y134, L137, and L138.

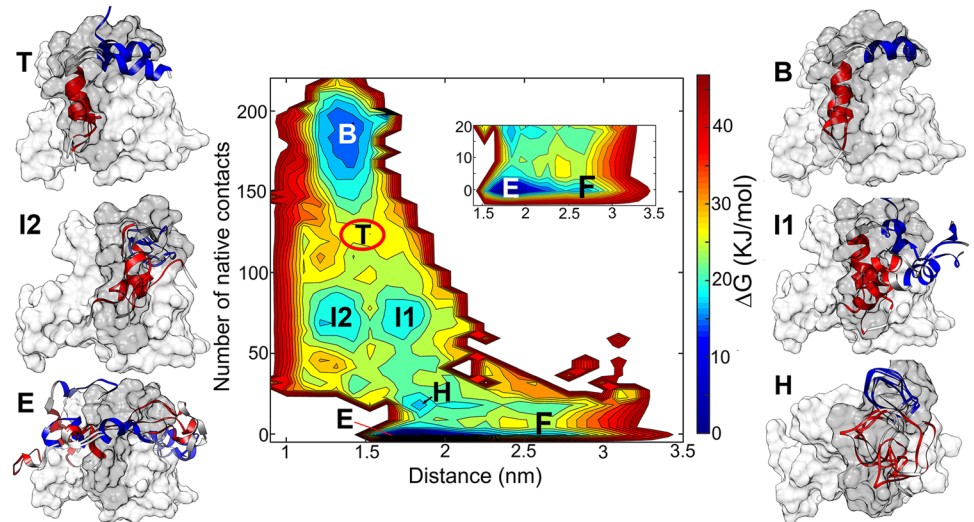

**Fig. 2 Free energy landscape (FEL) of pKID and KIX binding process.** The FEL is the function of the COM distance and native contacts number. Representative structures of the minima are given and overlapped with the experimental NMR structure. The NMR complex structure complex is shown in surface, and the pKID is colored in dark-gray and KIX is colored in gray. The simulated structures of pKID are represented in the cartoon style, and the $\alpha_A$ regions of simulated pKID are colored in blue, $\alpha_B$ regions are colored in red. The coordinates in PDB format of the representative structures is given in Supplementary Data 1.

FEL, i.e., the free state with the large distance and low contacts (the state marked by F), the encounter complex with distance close to 1.7 nm and the fraction of native contact Q close to 0 (marked by E), the intermediates with many native contacts formed (state I1 and I2) and the bound state with most of the native contact formed (state B). The structure properties of the important free energy minima are given in the following section.

**The free state.** In the free state, the pKID peptides are far away from KIX, the center of mass distances between pKID and KIX are larger than 25 Å. In the free-state conformations, neither native contacts nor non-native contacts are formed (Supplementary Fig. 2). The secondary structure composition of pKID in this state is similar to the apo-pKID, the helical contents on the $\alpha_A$ region is 52.0% and on $\alpha_B$ region is 10.2%.

**The encounter complex state.** Based on the $^1$H-$^{15}$N HSQC spectrum and $^{15}$N R$_2$ dispersion experiments[27], Sugase et al.

found ensemble of weak complex which fast exchanging with the free pKID, and these states were defined to be encounter complex. Similar to the experimental observation, the interactions between pKID and KIX in the encounter complex state determined by our simulations are dynamic. Multiple hotspot binding sites present on the KIX and pKID (Fig. 3a), including the hydrophobic interactions between residue Leu128 on pKID with Leu664 on KIX, residue Leu141 on pKID with Val635, Met639 and Leu652 on KIX. In addition, the interactions between the charged residues also contribute to the formation of the encounter complex, such as Arg135 on pKID and Glu648 on KIX, Asp140 on pKID, and Lys659 on KIX. It should be noticed that most of the interactions in the encounter complex are non-native contacts, which indicate the non-native interactions are major force to drive the formation of pKID-KIX encounter complex. On the other side, the phosphorylated serine (pSer133) would not direct contribute to the formation of encounter complex since there is no interactions between the pSer133 and the residues on KIX.

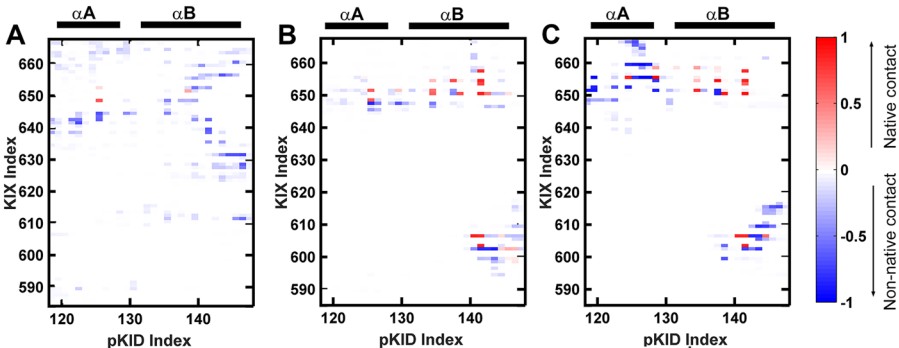

**Fig. 3 Contact maps between residues on pKID and KIX. a** Contact map between residues of pKID and KIX in the encounter complex (state E). **b** Contact map of intermediate 1 (state I1); **c** Contact map of intermediate 2 (state I2); In these contact map, red grid represents the native contacts are formed, blue grids are corresponding to the non-native contacts.

**Intermediates and hidden intermediate**. Two intermediates (I1 and I2) with similar native contact numbers but different COM distances were observed on the free energy landscape. Many residues of $\alpha_B$ regions are correctly anchoring with the native-binding sites on the KIX in both I1 and I2 intermediates, for example, the hydrophobic interactions between the residues Tyr134, Ile137, Leu138, Leu141 on pKID and the residues TyrY650, Ala654, Ile657, Tyr658 on KIX. Besides, the salt bridges interactions between residues Asp140 on pKID and Lys606 on KIX may also contribute to the binding. The residues on $\alpha_A$ region are more flexible and less contacts formed with KIX than residues in $\alpha_B$ region in the intermediates. The contacts between $\alpha_A$ region and KIX are different in I1 and I2, where Arg125 on pKID contact with Glu648 and His651 on KIX I1 (Fig. 3b), however, Arg124 mainly contact with Glu655 in intermediate I2 (Fig. 3c). The non-native contacts also contribute to the stabilization of the intermediates. However, more non-native interactions are formed in the intermediate I2, especially the residues on $\alpha_A$ region of I2 (Fig. 3c).

Although the native contact numbers are similar for intermediates I1 and I2, the secondary structure compositions are different in the two intermediates. In I1, the C-terminal of $\alpha_A$ are basically folded to helical structures (the helicity from 124 to 128 is close to 90%); the helicity of $\alpha_B$ in I1 is slightly lower than $\alpha_A$ (the helicity of these intermediates shown in Supplementary Fig. 3 and representative structures displayed in Fig. 2). This is consistent with the NMR characterization of the intermediate state[27] in which residues 124–128 in $\alpha_A$ nearly fully folded and the helicity of residues 133–138 and 141 in the $\alpha_B$ region is only about 70%. However, the pKID mainly adopt disordered structure in state I2, only several residues in $\alpha_B$ region have ~20% probability forming $\alpha$-helix[27]. By analyzing the FEL along $\alpha_A$ or $\alpha_B$ helicity and native contact number (Supplementary Fig. 4), we found that the structures of I2 are located on the off-pathway regions of the FEL.

In addition, a free energy minimum between the encounter complex and the intermediates was characterized. We denote the state as hidden intermediate (state H) since it has never been uncovered before. Compared with the encounter complex, some native contacts initially formed in state H. The contacts formed in this state might be corresponding to the primary driven force for the binding. The contact analysis showed that the intermolecular contacts between the aromatic residue Tyr658 on KIX and the hydrophobic residues Leu128 and Ile137 on pKID formed in state H, the contact probabilities are 80.3% and 62.1%, respectively. These residues are important to the pKID-KIX binding, which was proved in the single-residue mutagenesis experiment, the Y658A mutation would completely abrogate the complex

formation[16], the L128A and I137A mutation would increase the dissociation constant ($K_d$) over tenfold and two orders of magnitude, respectively[26].

**Fully bound state and free energy barrier**. There is a high energy barrier (state T) between the intermediate state and the final-binding state. The pKID further folding to helical structures in this state compared with state I1 and I2, especially for the $\alpha_B$ region, which is almost fully folded in state T. The native contacts on the $\alpha_B$ regions are almost fully formed, however, the contacts in the αA regions are partially formed with relatively low probability (Supplementary Fig. 2). The results mean that $\alpha_A$ haven't bound to the right position (binding sites) of KIX in the state T. It should be noticed that the native contacts between the phosphorylated serine on pKID and the residues Tyr658 and Lys662 on KIX are formed in this state, with the contact probability 66.8% and 64.2%, respectively, which might be the driving force for $\alpha_A$ region binding to the right position on KIX and the fully folding of $\alpha_A$.

In the fully bound state, the residues on pKID bound to the native-binding sites of KIX. The $\alpha_B$ region and the C-terminal of $\alpha_A$ region (residue 128–132) completely folded, the helicity on these regions are close to 100%. The helicity of N-terminal of $\alpha_A$ increase to 40%. The incomplete folding of N-terminus of $\alpha_A$ was also demonstrated by Dahal et al.[21], they found that the final bound complex of pKID-KIX is partially mobile, with $\alpha_A$ loosely bound to the KIX based on the kinetic experiments.

**The free energy landscape for KID-KIX binding**. Unlike the pKID, the unphosphorylated KID is hard to form stable complex with KIX. The binding affinity of unphosphorylated KID to KIX is about 100 times lower than the pKID[21]. To describe the binding behaviors of KID and KIX, the FEL as a function of KID-KIX COM distances and the fraction of native contacts ($Q$) was given in Fig. 4. The $Q$ values is corresponding to the native contact atoms in the structure of pKID-KIX complex. Compared with pKID, KID is hard to form stable complex with KIX since the high free energy in the high Q region. Interestingly, KID and KIX also form encounter complex and some intermediate states with very low native contacts. For the structures in the encounter complex state, the binding sites are variable and transient, which is similar to the state E in pKID binding to KIX. The contact map shows the interactions between KID and KIX in the intermediates were mainly happened in $\alpha_B$ region (Supplementary Fig. 5), in which the hydrophobic interactions might dominant the interactions. In the binding process of pKID and KIX, two intermediates with native contacts number close to 80 were observed.

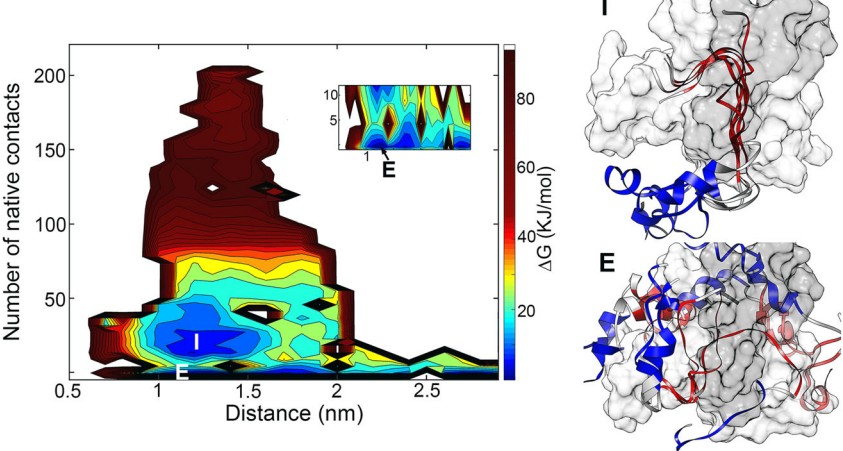

**Fig. 4 Free energy landscape (FEL) of KID and KIX binding process.** The FEL is the function of the KID-KIX distance and native contacts number, which formed in pKID-KIX. Representative structures of states E and I are shown. The structure of KID-KIX complex derived from pKID-KIX crystal structure and shown by surface in KID-KIX complex, KID color in dark-gray and KIX color in gray. The structures of KID in major states was shown in ribbon and $\alpha_A$ region color in blue, $\alpha_B$ region color in red. The coordinates in PDB format of the representative structures are given in Supplementary Data 2.

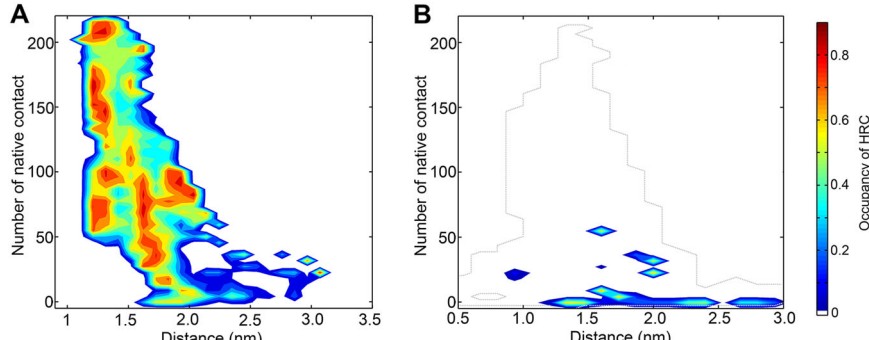

**Fig. 5 The HRC formation propensity. a** The formation probability of HRC as a function of the pKID-KIX distance and native contacts number which formed in pKID-KIX. **b** The formation probability of HRC as a function of the KID-KIX distance and native contacts number which formed in pKID-KIX. The formation of hydrophobic residue cluster (HRC) is defined by the contact atoms in residues L128, Y134, I137 and L138 larger than 15. Only the states with the free energy lower than 20 kJ/mol are given. The profile of free energy landscape of KID-KIX binding is given by dashed-line.

However, it is hard to form the similar intermediates in KID-KIX system. Besides, KID is basically incompletely folded and unstructured in all the states interact with KIX. Our results indicate that KID is transiently contacted with KIX and difficult to form stable and ordered complex.

**The hydrophobic residue cluster formed in the binding process of pKID**. The mutagenesis and kinetic experiments demonstrated that the hydrophobic residues Leu128, Tyr134, Ile137, Leu138, and Leu141 are important in the pKID-KIX binding. Based on our simulations, we found the hydrophobic interactions related to these residues formed prior to the formation of pKID-KIX binding intermediates. On the other hand, the interactions between these residues are absent in the unphosphorylated KID and the binding and folding process of KID and KIX would not proceed after the encountering complex. The results demonstrate the interactions between the hydrophobic residues Leu128, Tyr134, Ile137, Leu138, and Leu141 play important roles in stabilizing and guiding pKID-KIX binding.

By analyzing the structure properties of conformations in the binding process, we found the special structure pattern, i.e., the HRC, also appeared in pKID. The HRC (the number of contact heavy atoms in the side-chain of Leu128, Tyr134, Ile137, and Leu138 are larger than 15) formation probability were projected on the pKID-KIX and KID-KIX binding FEL (Fig. 5a and b), respectively. It can be seen that the average probability of HRC in the unphosphorylated KID is lower than 10%, indicate that the HRC structures are basically absent in the KID. On the other side, the conformations of pKID prefer to form the HRC, especially the conformations in the hidden state H. As the state H plays important role in the binding process of pKID and KIX, the formation of HRC and its anchoring to the binding site on KIX might provide the initial force of correct binding. We inferred that the HRC amplifies the hydrophobic interaction ability of pKID and facilitate the pKID to search for favorable binding sites on KIX and finally fold to the bound state.

## Discussion

**Phosphorylation promotes the formation of hydrophobic residue cluster.** Phosphorylation on the Ser133 of KID dramatically increase the binding affinity of the peptide to KIX, however, the underlying mechanism is still unclear. The pSer133 is contact with the residues Lys662 and Tyr658 of KIX in the experimental complex structure, and the mutagenesis experiment shows that Y658A mutation on KIX completely abrogate the complex formation[16]. Therefore, the hydrogen bond interaction between the pSer133 and the Tyr658 might great contribute to the stabilization of the pKID-KIX complex. However, the kinetic

experiments show the binding Φ-values of residues around pSer133 are low or negative, indicates the phosphorylation has litter effects in initiating the binding of pKID to KIX. Our simulations results are consistent with the experimental observations. Barely contacts between pSer133 and residues in KIX were formed at the early stage of the binding process, i.e., the free state, the encounter complex and the intermediates. The minimum distance between pSer133 and residues on KIX is larger than 10 Å in the encounter complex and intermediate states (Supplementary Fig. 6). The NMR experiment found a large chemical shift change on pSer133 caused by the encounter complex formation (the chemical shift change on backbone amide of pSer133 is 0.34 ppm). The chemical shifts of pSer133 also changed dramatically in our simulation, i.e., the chemical shift difference of backbone amide of pSer133 in the encounter complex and free state is about 0.2 ppm. The results indicate that even no obvious interactions formed between pSer133 and KIX, the local environment change when approaching KIX may induce the chemical shift perturbation on pSer133. However, it can't be excluded that the alternative pathways were observed in the NMR experiment which would induce larger chemical shift perturbation on pSer133.

On the other side, the phosphorylation on Ser133 promote the formation of HRC in pKID. Upon the phosphorylation, the phosphate group prefer to form hydrogen bonds with its nearby positively charged residues, such as Arg124, Arg125, Arg 130, Arg 131, and Lys136 in pKID. The phosphorylated group on pSer133 has high propensity to contact with the positively charged residues Arg131 (49.3%) and Lys136 (57.2%) in free pKID. In comparison, the contact probability of Ser133 with Arg131 and Lys136 in KID are 16.3% and 10.4%, respectively. (Supplementary Fig. 7). As the hydrophobic residues of the HRC are around pSer133 and the positively charged residues, the interactions would facilitate the approaching of these hydrophobic residues and the formation of HRC.

**Hydrophobic residue cluster in the binding process.** The HRC amplifies the hydrophobic interaction ability of KID and facilitates the searching of appropriate binding sites on KIX. In order to identify the interaction partner on KIX with the HRC, we calculated the interactions between the residues in HRC and residues on KIX in the hidden state. The results show that the HRC prefers to contact with the C-terminal helix (helix 3) of KIX; in particular, there are high interaction probabilities between residues Leu128 and Ile137 on pKID with Tyr658 on KIX (Fig. 6). The contact probability between L128 or I137 and Y658 are larger than 0.6 in the hidden state H, which is corresponding to the initial stage of the binding process. The results demonstrated

that the interactions between the HRC and residue Y658 play important role in the binding process. In fact, the important roles of Y658 in the binding process was observed by the mutagenesis experiments, Radhakrishnan et al. found that the single mutation Y658A on KIX would completely abolish the complex formation[16].

**Flexible conformation selection in the IDP binding.** The HRC structures are observed in both of the free pKID and the pKID-KIX complex. More than 40% conformations with contact number larger than 15 in the free state pKID. The abundant amount of HRC structures of free pKID allow the direct binding of these conformations with KIX. In fact, the HRC structures of pKID in the hidden intermediate and free-state pKID are similar (Supplementary Fig. 8). As a result, we conclude that the binding and folding of pKID with KIX follows a flexible conformational selection mechanism, i.e., some pre-formed structure patterns on the free IDP is required for the effective binding (the HRC structure on pKID in the case of pKID-KIX binding). However, unlike the solid conformation selection model, which is common in the binding of enzyme and substrates, only some local structure features are required. There is no need to sample the pre-formed structure with the ordered final configuration. In the pKID-KIX binding process, the structures of regions except the HRC are distinct with each other (Supplementary Fig. 9). This binding model proposed here allows the binding of IDP with high efficiency.

**The folding steps of $\alpha_A$ and $\alpha_B$.** One major debate about the pKID-KIX binding is the order of $\alpha_A$ and $\alpha_B$ folding, i.e., which part complete the folding firstly. The kinetic experiment shows that the binding and folding of $\alpha_B$ region prior to the binding and folding of $\alpha_A$. Nevertheless, NMR results show that the chemical shifts differences of the $\alpha_B$ region in the intermediates and bound state are larger than the $\alpha_A$ region, indicate the $\alpha_B$ region are less folded than $\alpha_A$ in the intermediates.

The controversy can be explained and clarified by our simulations. By analyzing the residue helicity of intermediates (Supplementary Fig. 3), we found that the helix in the C-terminal region of $\alpha_A$ are basically folded in intermediate I1. The average helicity in this region (residue 124–128) is about 84%. The average helicity on the $\alpha_B$ region (residue 134–144) is near 40%, which is consistent with the NMR experiments. On the other side, both the $\alpha_A$ and $\alpha_B$ are basically finished the folding in the transition state. However, the $\alpha_A$ does not bind to the correct sites on KIX. The distributions of $\alpha_A$–$\alpha_B$ angles in different states are given in Supplementary Fig. 9. The incorrect position of $\alpha_A$ might be the reason why the Φ-values of $\alpha_A$ region are much smaller

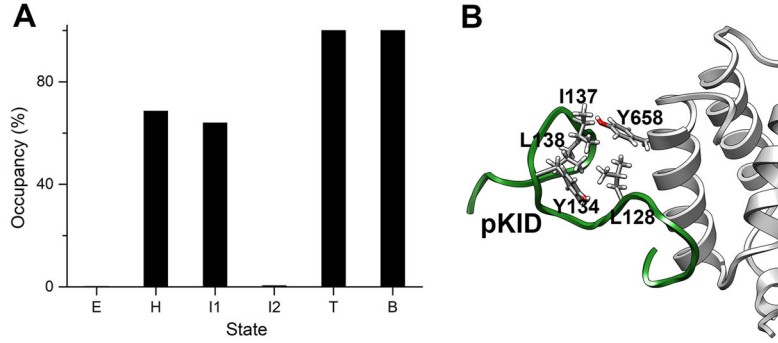

**Fig. 6 The interactions between Y658 and HRC residues. a** The contact probability between the hydrophobic residues in HRC with the Y658 (on KIX) in the encounter complex (state E), the hidden state (state H), intermediates (states I1 and I2), the transition state (state T) and the bound state (state B). **b** The representative structure of state H of pKID and KIX binding.

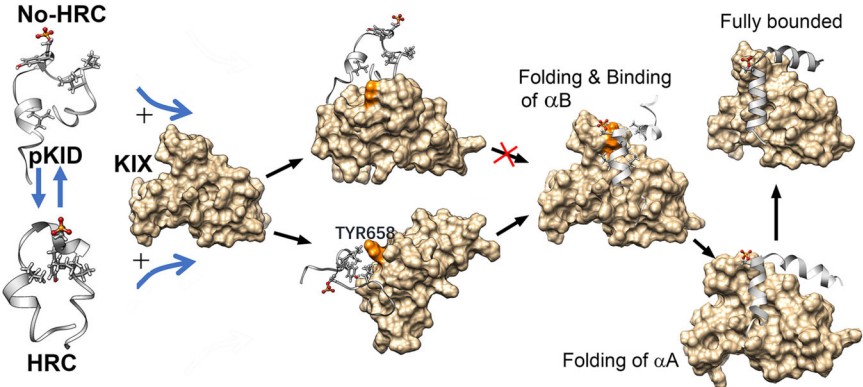

**Fig. 7 The model of coupled folding and binding process of pKID and KIX.** Before the encountering with KIX, the free pKID adopts flexible conformations and quick transiting between the conformations with HRC formed and the conformations without HRC. The pKID dynamically contact with KIX on multiple "hot-pot" binding sites to form the encounter complex, however, the pro-formed HRC structure would promote the correct binding of pKID to the Tyr658 on KIX. The accurate docking further promote the binding of $\alpha_B$ region. The binding with KIX also would induce the folding of C-terminal region of $\alpha_A$ and the $\alpha_B$ region of pKID. After that, the N-terminus of $\alpha_A$ would complete the folding. The folded $\alpha_A$ of pKID would rotate to the correct positions on KIX.

than the $\alpha_B$ region. Interestingly, the structures with incorrect $\alpha_A$–$\alpha_B$ angles were also observed in a recent computational study, though they proposed these structures are in "misfolding" state[29].

**The model of pKID-KIX binding process**. A model is proposed to describe the binding and folding process of pKID-KIX based on the observations in this work (Fig. 7). In the free state, pKID adopts flexible conformations and quick transiting between these conformations. Due to the electrostatic interactions between pSer133 and positively charged residues on pKID, the hydrophobic residues around the positively charged residues incline to assemble together and form the HRC. Without the HRC structural pattern, pKID dynamic contact with KIX on multiple "hot-pot" binding sites to form the encounter complex, however, the pro-formed HRC structure would promote the pKID correct bind to the Tyr658 on KIX. The HRC is critical in the forming of intermediates. The folding and binding of pKID should obey the flexible conformational selection mechanism, which means the structure is high changeable except the HRC structure. The C-terminal region of $\alpha_A$ firstly complete the folding in the intermediates, meanwhile, the $\alpha_B$ region is more flexible and the N-terminal of $\alpha_A$ is unfolded. After the folding and binding of $\alpha_B$ region, the N-terminus of $\alpha_A$ complete folding. The folded $\alpha_A$ would rotate to the correct positions and finally bound with KIX by crossing the energy barrier.

## Conclusions

The IDPs are usually binding with partners to carry out their biological functions. It is still ambiguous why IDP could adopt high specificity and efficient ability in the binding. In this study, the binding mechanism of pKID to KIX, which is regulated by phosphorylation on the serine in the central region of KID, was investigated by the computational simulations. The structure properties of free pKID and KID, as well as the binding process of pKID with KIX were characterized. The enhanced sampling results show that both free-state pKID and KID are disordered except with some transient helical structures on them. Similar to the experimental observation, no obvious differences were observed on the secondary structure composition of the pKID and KID. However, more hydrophobic interactions formed in the pKID, which promote the formation of the special HRC. Based on the binding free energy surface of pKID and KIX, the critical sites and important intermediates on the binding process were characterized. The HRC structures were also observed on the pKID in the present of KIX. Although both of KID and pKID would form

encounter complex with KIX, only the structures with HRC preformed in free pKID would bind to the correct binding sites on KIX and further fold to the final structure. The binding mechanism of the intrinsically disordered pKID follows a flexible conformational selection mechanism, i.e., the substrate protein specifically binding with the IDP with some special locally structures, meanwhile, most of the rest regions on the IDP are flexible and dynamic. The flexible conformational selection model proposed in this work give the explanation of the high specific and efficient of IDP binding. The binding mechanism proposed in this work provide new insights in the protein dynamic interactions and phosphorylation regulation.

## Methods

**System setup**. The structures of free pKID and KID were built based on the experimental structure of pKID-KIX complex (PDB ID: 1KDX[16]). The phosphorylated pS133 was mutated back to the serine to build the structure of KID. The N- and C- terminus of pKID and KID were capped with acetyl (ACE) and amine (NH2) groups, respectively. 4324 and 5546 water molecules were added to solvating pKID and KID, respectively. Sodium ions and chlorine ions were added to neutralize the systems and the final concentrations of the sodium chloride were set to be 100 mM. The amber99SB-ILDN force field[30] was employed for proteins, TIP3P model[31] was used for water molecules. The force field parameters of phosphoserine were taken from Steinbrecher et al.'s work[32]. The unbiased molecular dynamics simulations were performed to equilibrate the conformations of free pKID and KID in aqueous solution. The systems were energy-minimized for 5000 steps using the steepest descent method. Then the NVT simulations with position constraints applied on protein heavy atoms and NPT simulation were performed to equilibrate the systems. All bonds length were constrained with LINCS algorithm[33]. Isotropic scheme was utilized to couple the lateral and perpendicular pressures separately. The Particle-Mesh Ewald method was employed to calculate long-range electrostatics with a cutoff of 10 Å. The temperature was kept at 300 K with V-rescale method[34] and the pressure were coupled by Parrinello–Rahman barostat[35]. 500 ns production runs were performed with 2-fs time step by GROMACS5.1.4[36,37].

**The well-tempered metadynamics combined with parallel tempering**. The parallel tempering simulations combined with well-tempered ensemble (PTMetaD-WTE) method[38–40] was employed to obtain the structure ensemble of free pKID and KID. The initial structures of pKID and KID in the PTMetaD-WTE were obtained from the last snapshots of 500 ns unbiased molecular dynamics simulations. Twelve replicas were simulated spanning the temperature range of 288–508 K. The temperature distribution of these replicas was chosen according to the Ref. [40]. The PTMetaD-WTE simulations were implemented in two-step scheme. First, the parallel tempered simulations in the well-tempered ensemble (PT-WTE) were employed to sample the conformations by using the potential energy as the collective variable. The bias factor $\gamma$ was set to be 30. The height of initial bias energy is 1.0 kJ mol$^{-1}$ and the width is 300 kJ mol$^{-1}$. Exchange of configurations between adjacent replica was attempted every 150 fs. After the 30 ns simulations on each replica, the height of the bias energy decreased to the value close to zero and the exchange acceptance probability between adjacent replicas are about 0.3. The average potential energy in PT-WTE simulations remains close to the canonical

value with large fluctuations. In the second step, simulations in all replicas were performed with a static energy bias in the potential energy landscape which constructed in PT-WTE. The history-dependent energy bias is added on two collective variables to enhance the sampling of the structure on $\alpha_A$ and $\alpha_B$ regions of KID. The description and definition of the CVs are given in the supporting information. The PTMetaD-WTE simulation on each replica is 300 ns and the total simulation time is 3.6 μs. All simulations were performed using GROMACS5.1.4 and PLUMED-2.1 plugin[41]. The average exchange acceptance ratio between the replicas is 0.35 for pKID and 0.39 for KID, respectively.

**Bias-exchange metadynamics (BE-MetaD)**. BE-MetaD combined the ideas of replica exchange and metadynamics[42,43], the simulations are exchanged in different replicas, which could be present by different collective variables. In this way, a large number of different variables can be sampled simultaneously. In this work, BE-MetaD simulation performed in well-tempered ensemble[39]. The initial structure was built based on the experimental complex structure (PDB ID: 1KDX). 10092 and 9746 water molecules were added to solvating pKID-KIX and KID-KIX complexes, respectively. Sodion and chloridion were added to neutralize the charged systems and the final concentrations of the sodium chloride were set to be 100 mM. Four biased replicas run along with four CVs for the BE-MetaD simulations, the descriptions and definitions of the CVs are given in supporting information. The exchanges between the replicas were attempted every 4 ps. 450 ns simulation was performed on each replica, and totally 1.8 μs for each system. The convergence of these simulations is shown in Supplementary Fig. 10.

**Analysis**. Secondary structures were assigned using the STRIDE module in VMD[44,45]. Chemical shifts were calculated by Shfitx2[28]. Free energy landscapes as a function of CVs for pKID/KID-KIX system were calculated by weighted histogram analysis method by using the METAGUI program[46]. The reweighted value of the observable property $O$ were calculated based on the estimated free energies as following[43]:

$$\langle O \rangle = \sum_\alpha O_\alpha \exp(-F_\alpha/k_B T) / \sum_\alpha \exp(-F_\alpha/k_B T), \quad (1)$$

where $F_\alpha$ is the free energy of cluster α, $k_B$ is the Boltzmann constant, $T$ is the system temperature, the sums run over all the clusters of the free energy profile. $O_\alpha$ is the average value of $O$ in cluster α.

## Data availability

The data generated in this study are available from corresponding author upon request.

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

## Acknowledgements
This work is supported by the National Natural Funding of Science (21773298).

## Author contributions
D.M. conceived and supervised the project, L.N. performed the MD simulations and analyzed the data. G.Y. contributed to analysis. L.N., G.Y., N.S., and D.M. contributed to the scientific discussion. D.M., and L.N. wrote the paper.

## Competing interests
The authors declare no competing interests.
