## [Peer Review File · Communications Chemistry]

Reviewers' comments:

Reviewer #1 (Remarks to the Author):

Liu et al. computationally studied the process of an intrinsically disordered protein, the phosphorylated pKID domain of CREB, binding to the KIX domain of CBP. The authors conducted all-atom molecular dynamic simulations, called bias-exchanged meta dynamics (BE-MetaD) simulation, of the free state of pKID and KID as well as the binding process of pKID and KID to KIX. They found that phosphorylation of Ser133 of KID induces the formation of "special hydrophobic residue clusters (HRC)", which promotes the binding of pKID to the correct binding site in KIX. Based on the results, the authors proposed the flexible conformational selection mechanism to explain the high specificity and efficiency of IDPs.

For the free state of KID and pKID, they excellently reproduced previous experimental data, that is, ¹³Ca and ¹Hα chemical shifts predicted from the simulated structures are in good agreement with the experimental ones. For the binding process, however, their result is inconsistent with the previous NMR data; the authors found that there are no interactions between pSer133 and KIX in the encounter complex, but Sugase et al showed a large chemical shift change of pSer133 that is caused by the encounter complex formation (Nature 447, 1021 (2007)). This discrepancy infers that pKID binds to KIX in a different pathway in the simulation from experimentally characterized one. This point would need to be revised. Furthermore, there are some issues:

This manuscript is written in poor English. I would suggest that the manuscript is carefully reviewed by an experienced editor whose first language is English and who specializes in editing.

The reference numbers are not organized, and some sentences require appropriate citation. For example, "A recent kinetic experiment shows that the binding and folding of αB region prior to the binding and folding of αA." on page 4. "Interestingly, the structures with incorrect αA-αB angles were also observed in a recent computational study, though they proposed these structures are in "misfolding" state." on page 15.

On page 3, "the detailed mechanism about the regulation on the structures and the interactions of IDPs are still poorly understood,"

This seems overstatement because there are so many studies of IDPs.

The KID construct used in the NMR studies are not capped at the N- or C-termini. The electrostatic property should be changed by these caps.

On page 8, "In the free-state conformations, neither native contacts nor non-native contacts are formed (Figure S3)."

There is no contact map for the free state in Figure S3.

On page 13, "The results show that the HRC contact with the C-terminal helix (helix 4) of KIX," "helix 4" should be "helix 3".

On pages 13,14 "The interactions between the HRC and residue Y658 might be the initial driven force for the correct binding process. Actually, the mutagenesis experimental show that single mutation Y658A on KIX would dramatically decrease the binding affinity of pKID, which is consistent with our observations."

Y658 in KIX is an important residue for pKID recognition, but only from the mutation study, it cannot be proved that Y658 is involved in the initial state of the interaction between pKID and KIX.

In the legend of Figure 2, "the crystal structure of pKID and KIX complex was shown in surface," "crystal" should be "solution".

Reviewer #2 (Remarks to the Author):

The work by Duan and coworkers describes the conformational selection mechanism regulated by phosphorylation of serine residue of an IDP (KID) and the formation of the macromolecular complex (KID-KIX) coupled to the folding event.

The results presented are highly interesting, the highlight on intermediates I1 and I2 is novel, the work provides new insights in the coupling post-translational modifications to the protein dynamics and folding, and the choice of methodology is adequate. Before the work can be published, however, the authors should address the following comments:

1. In Abstract: "Most of the phosphorylation happen on the intrinsically disordered proteins (IDPs)." How was it quantified? I am not convinced this statement is true. Phosphorylation is ubiquitous post-translational modification, occurring in structured, globular proteins and IDPs alike.

2. How were the parameters for pSer obtained? This should be described in Methods.

3. Choice of the force field: Amber99SB-ILDN is known to bias the simulations towards the structured and collapsed configurations. Could authors explain the choice of the force field?

4. pp. 5 and 6: "The initial structure of pKID/KID in PTMetaD-WTE simulation was obtained from the unbiased molecular dynamics." – how?

5. p.6: in Be-MetaD – what where the collective variables chosen? The protocol needs to be described more in detail.

6. Why is chemical shifts calculation not described in methods?

7. Language: before publishing, the manuscript should be edited, ideally by a native English speaker. There are numerous typos, grammar, style, punctuation and spelling mistakes. To give only a few examples: "the predicted chemical shifts are agree well with the NMR measurement values"; "The helix occupancy on both of the α A region of pKID and KID are about 50%"; "terminuses"; "Barely contacts between pSer133 and residues in KIX were formed at the early stage of the binding process"; "positive charged residues"; and many, many more.

8. Also, the authors should stick to either one letter, or three letter amino acid code, not to mix them.

Responses to Reviewers:

We wish to thank the referees for a careful reading of our manuscript. The referees provided us valuable comments and advices. We revised our manuscripts accordingly. In the following the referees' comments are in italic font and colored in blue, our responses are in regular font.

Reviewer #1:

1. Liu et al. computationally studied the process of an intrinsically disordered protein, the phosphorylated pKID domain of CREB, binding to the KIX domain of CBP. The authors conducted all-atom molecular dynamic simulations, called bias-exchanged meta dynamics (BE-MetaD) simulation, of the free state of pKID and KID as well as the binding process of pKID and KID to KIX. They found that phosphorylation of Ser133 of KID induces the formation of "special hydrophobic residue clusters (HRC)", which promotes the binding of pKID to the correct binding site in KIX. Based on the results, the authors proposed the flexible conformational selection mechanism to explain the high specificity and efficiency of IDPs. For the free state of KID and pKID, they excellently reproduced previous experimental data, that is, ^{13}C and ^1H chemical shifts predicted from the simulated structures are in good agreement with the experimental ones.

For the binding process, however, their result is inconsistent with the previous NMR data; the authors found that there are no interactions between pSer133 and KIX in the encounter complex, but Sugase et al showed a large chemical shift change of pSer133 that is caused by the encounter complex formation (Nature 447, 1021 (2007)). This discrepancy infers that pKID binds to KIX in a different pathway in the simulation from experimentally characterized one. This point would need to be revised.

Reply: We would like to thank the reviewer for the valuable comments. Actually, our simulation results about the binding processes are consistent with the previous NMR data in many aspects. First of all, the bind process of pKID and KIX follows a four-site exchange model based on the NMR studies. All the four stages were clearly captured on the free energy landscape obtained by our simulations, including the free energy minima of free state, encounter complex, the intermediate and the bounded state. Second, we found that the helix in the C-terminal region of α_A are folded in the on-pathway intermediate state, meanwhile, the α_B is partially folded. The result is consistent with the NMR experiment, which shows that the chemical shifts differences of the α_B region in the intermediates and bounded state are larger than the α_A region and indicates the α_B region are less folded than α_A in the intermediates. Besides, similar to the NMR experimental observations, the chemical shift of pSer133 also changed largely in the encounter complex in our simulations. We agree with the reviewer's comment which we can't exclude the possibility of different

pathways were characterized by the simulation and the NMR experiments. Therefore, we revised the related contents and add the following discussion:

“The NMR experiment found a large chemical shift change on pSer133 caused by the encounter complex formation (the chemical shift change on backbone amide of pSer133 is 0.34 ppm). The chemical shifts of pSer133 also changed dramatically in our simulation, i.e. the chemical shift difference of backbone amide of pSer133 in the encounter complex and free state is about 0.2 ppm. The results indicate that even no obvious interactions formed between pSer133 and KIX, the local environment change when approaching KIX may induce the chemical shift perturbation on pSer133. However, it can't be excluded that the alternative pathways were observed in the NMR experiment which would induce larger chemical shift perturbation on pSer133.”

2. This manuscript is written in poor English. I would suggest that the manuscript is carefully reviewed by an experienced editor whose first language is English and who specializes in editing.

Reply: We thank the reviewer for the careful reading of our manuscript. We have revised the paper carefully and asked some native speakers to improve the English expressions.

3. The reference numbers are not organized, and some sentences require appropriate citation. For example, “A recent kinetic experiment shows that the binding and folding of αB region prior to the binding and folding of αA .” on page 4. “Interestingly, the structures with incorrect αA - αB angles were also observed in a recent computational study, though they proposed these structures are in “misfolding” state.” on page 15.

Reply: We thank the reviewer for the useful suggestion. We carefully checked the reference citations and added the missing citations to the text.

4. On page 3, “the detailed mechanism about the regulation on the structures and the interactions of IDPs are still poorly understood,” This seems overstatement because there are so many studies of IDPs.

Reply: We thank the reviewer for the valuable suggestion. We rephrased the sentence to:

“However, the detailed mechanism about the phosphorylation regulation on the structures and the interactions of IDPs remain elusive, which greatly impede the understanding of IDP function and the searching for “druggable” IDP targets.”

5. The KID construct used in the NMR studies are not capped at the N- or C-termini. The electrostatic property should be changed by these caps.

Reply: The N- and C-termini of KID were capped with acetyl (Ace) and amine (NH₂) groups in our simulations to mimic the full-length protein. Since KID is a part of CREB protein, we capped the two terminals of KID to mimic the connections of them with other residues. The Ace and NH₂ we utilized here are two widely used terminus in the MD simulations. On the other side, based on the NMR experiment, the chemical shifts of two terminus are basically unchanged during the binding process (Figure 2 in Sugas et al's 2007 Nature), which indicate there is no interaction with KIX and no large structure changes of the terminal residues during the binding process. Therefore, the caps on the two termini of pKID have little influence on the pKID binding process.

6. On page 8, "In the free-state conformations, neither native contacts nor non-native contacts are formed (Figure S3)." There is no contact map for the free state in Figure S3.

Reply: We thank the reviewer for the careful reading of our manuscript. The contact map of free-state was added to the supporting information (Supplement Figure 3D).

7. On page 13, "The results show that the HRC contact with the C-terminal helix (helix 4) of KIX," "helix 4" should be "helix 3".

Reply: We thank the reviewer for the careful reading of our manuscript. We revised "helix 4" to "helix 3".

8. On pages 13,14 "The interactions between the HRC and residue Y658 might be the initial driven force for the correct binding process. Actually, the mutagenesis experimental show that single mutation Y658A on KIX would dramatically decrease the binding affinity of pKID, which is consistent with our observations." Y658 in KIX is an important residue for pKID recognition, but only from the mutation study, it cannot be proved that Y658 is involved in the initial state of the interaction between pKID and KIX.

Reply: We thank the reviewer for the valuable suggestion. The interactions between residues L128/I137 on pKID with the residue Y658 on KIX in different state were analyzed (Figure 6). The contact probability between L128/I137 and Y658 are larger than 0.6 in the hidden state H, which is corresponding to the initial stage of the binding process. That is why we proposed that residue Y658 may play important roles from the initial binding process. We agree with the reviewer about the mutagenesis experiment is not enough to prove the Y658 is involved in the initial stage, therefore, we revised the corresponding sentences to:

“The results demonstrated that the interactions between the HRC and residue Y658 play important role in the binding process. In fact, the important roles of Y658 in the binding process was observed by the mutagenesis experiments, Radhakrishnan et al. found that the single mutation Y658A on KIX would completely abolish the complex formation.^{[46]”}

9. In the legend of Figure 2, “the crystal structure of pKID and KIX complex was shown in surface,” “crystal” should be “solution”.

Reply: We thank the reviewer for the useful suggestion. We revised “crystal” to “solution” in the legend of Figure 2.

Reviewer #2

1. In Abstract: “Most of the phosphorylation happen on the intrinsically disordered proteins (IDPs).” How was it quantified? I am not convinced this statement is true. Phosphorylation is ubiquitous post-translational modification, occurring in structured, globular proteins and IDPs alike.

Reply: We thank the reviewer for the useful comments. Phosphorylation sites are frequently associated with IDPs or IDRs. By investigating more than 1500 experimentally determined phosphorylation sites in eukaryotic proteins, Iakoucheva *et al.* found that phosphorylation commonly occurs within intrinsically disordered protein regions. (Nucleic Acids Res 2004, 32(3): 1037-1049). The above point was widely adopted by researchers, such as Bah and Forman-Kay said in a review paper: “At least a third of eukaryotic proteins may be phosphorylated, and most phosphorylation sites are within intrinsically disordered regions. (Bah and Forman-Kay, J. Biol. Chem., 2016, 291:6696). And Wright et al proposed “Phosphorylation sites are located predominantly in IDRs...” (Wright and Dyson, Nat. Rev. Mol. Cell Biol., 2014, 16:18). We deleted the sentence in the Abstract due to the word number limitation of abstract, and revised the related sentence in the Introduction to:

“Most of the phosphorylation sites are located in the intrinsically disordered proteins (IDPs) or intrinsically disordered regions (IDRs).^{[4-5]”}

2. How were the parameters for pSer obtained? This should be described in Methods.

Reply: The parameters obtained from Steinbrecher et al’s work. And the description of parameters was added to the *Methods*:

“The force field parameters of phosphoserine were taken from Steinbrecher *et al*’s work [29]”

3. Choice of the force field: Amber99SB-ILDN is known to bias the simulations towards the structured and collapsed configurations. Could authors explain the choice of the force field?

Reply: We thank the reviewer for the valuable comments. Amber99SB-ILDN is a widely-used and well-tested force field. It was thought the force field bias the simulations towards the structured configurations, However, we found the force field could accurately characterize the structure properties of free pKID and KID, and is also good agreement with many experiment observations of pKID-KIX binding process. We believe that is because pKID and KID are not fully disordered protein, there are two transient helices on them even in the free state. On the other side, the force field specifically for the disordered proteins might over sample the flexible configurations. Based on this, we think Amber99SB-ILDN force field is an appropriate force field for our simulations.

4. pp. 5 and 6: “The initial structure of pKID/KID in PTMetaD-WTE simulation was obtained from the unbiased molecular dynamics.” – how?

Reply: We thank the reviewer for the valuable comments. The initial structures of free pKID were extracted from the experimental structure of pKID-KIX complex (PDB ID: 1KDX^[26]). The phosphorylated Ser133 was mutated to the serine to build the structure of KID. To relax the experiment structures, 500-ns unbiased molecular dynamics simulation was performed on pKID and KID, respectively. Then the last snapshots of the unbiased trajectories were utilized as the initial structures of PTMetaD-WTE simulation. We added the following contents to Methods:

“The initial structures of pKID and KID in the PTMetaD-WTE were obtained from the last snapshots of 500-ns unbiased molecular dynamics simulations.”

5. p.6: in Be-MetaD – what where the collective variables chosen? The protocol needs to be described more in detail.

Reply: We thank the reviewer for the valuable suggestions. The detailed descriptions of the CVs employed in PTMetaD-WTE and BE-MetaD were given in the supporting information (page 2-3).

6. Why is chemical shifts calculation not described in methods?

Reply: We thank the reviewer for the useful suggestions. We added the analysis details in *Methods* on page 7:

“**Analysis.** Secondary structures were assigned using the STRIDE module in VMD software.^[40-41] Chemical shifts were calculated by Shfitx2.^[42] Free energy landscapes as a function of CVs for pKID/KID-KIX system were calculated by weighted histogram

analysis method(WHAM) by using the METAGUI program.^[43] The reweighted value of the observable property O were calculated based on the estimated free energies as following^[43]:

$$\langle O \rangle = \sum_{\alpha} O_{\alpha} \exp(-F_{\alpha}/k_B T) / \sum_{\alpha} \exp(-F_{\alpha}/k_B T) \quad (1)$$

where F_{α} is the free energy of cluster α , k_B is the Boltzmann constant, T is the system temperature, the sums run over all the clusters of the free energy profile. O_{α} is the average value of O in cluster α .”

7. Language: before publishing, the manuscript should be edited, ideally by a native English speaker. There are numerous typos, grammar, style, punctuation and spelling mistakes. To give only a few examples: “the predicted chemical shifts are agree well with the NMR measurement values”; “The helix occupancy on both of the αA region of pKID and KID are about 50%”; “terminuses”; “Barely contacts between pSer133 and residues in KIX were formed at the early stage of the binding process”; “positive charged residues”; and many, many more.

Reply: We thank the reviewer for the careful reading of our manuscript. We have revised the paper carefully and asked some native speakers to improve the English expressions. The mistakes mentioned above were corrected:

“the predicted chemical shifts are agree well with the NMR measurement values” to “the predicted chemical shifts are agreed well with the NMR measurements” .

“The helix occupancy on both of the αA region of pKID and KID are about 50%” to “The helicity of α_A are about 50% in both pKID and KID”.

“terminuses” to “terminals”.

“Barely contacts between pSer133 and residues in KIX were formed at the early stage of the binding process” to “Barely contacts were formed between pSer133 and residues in KIX at the early stage of the binding process” .

“positive charged residues” to “positively charged residues”.

8. Also, the authors should stick to either one letter, or three letter amino acid code, not to mix them. While the work is extensive in terms of computations, the presentation is somewhat poor, below standards and with countless grammatical errors and even repetitions of some sentences (e.g. see Page 5).

Reply: We thank the reviewer for the useful suggestion. We have unified the amino acid code to three letter code. And we carefully revised the English expressions and asked some native speaker to correct the grammatical errors.

REVIEWERS' COMMENTS:

Reviewer #1 (Remarks to the Author):

All my concerns have been addressed by the changes in the revised manuscript.

Reviewer #2 (Remarks to the Author):

The authors did a very good job in addressing all the comments and suggestions. After the revisions I am happy to recommend the manuscript to be published.